# EGCG-Derivative G28 Shows High Efficacy Inhibiting the Mammosphere-Forming Capacity of Sensitive and Resistant TNBC Models

**DOI:** 10.3390/molecules24061027

**Published:** 2019-03-15

**Authors:** Ariadna Giró-Perafita, Marc Rabionet, Marta Planas, Lidia Feliu, Joaquim Ciurana, Santiago Ruiz-Martínez, Teresa Puig

**Affiliations:** 1Perlmutter Cancer Center, NYU School of Medicine, 522 First Avenue, Smilow Research Building, Room 1104, New York, NY 10016, USA; ariadna.giroperafita@nyulangone.org; 2New Therapeutic Targets Laboratory (TargetsLab)-Oncology Unit, Department of Medical Sciences, Faculty of Medicine, University of Girona, Emili Grahit 77, 17003 Girona, Spain; m.rabionet@udg.edu; 3Product, Process and Production Engineering Research Group (GREP), Department of Mechanical Engineering and Industrial Construction, University of Girona, Maria Aurèlia Capmany 61, 17003 Girona, Spain; quim.ciurana@udg.edu; 4Laboratori d’Innovació en Processos i Productes de Síntesi Orgànica (LIPPSO), Department of Chemistry, University of Girona, Maria Aurèlia Capmany 69, 17003 Girona, Spain; marta.planas@udg.edu (M.P.); lidia.feliu@udg.edu (L.F.)

**Keywords:** FASN inhibition, triple-negative breast cancer, cancer stem cells, EGCG, G28, polyphenolic compound, fatty acid metabolism

## Abstract

Recent studies showed that Fatty Acid Synthase (FASN), a lipogenic enzyme overexpressed in several carcinomas, plays an important role in drug resistance. Furthermore, the enrichment of Breast Cancer Stem Cell (BCSC) features has been found in breast tumors that progressed after chemotherapy. Hence, we used the triple negative breast cancer (TNBC) cell line MDA-MB-231 (231) to evaluate the FASN and BCSC population role in resistance acquisition to chemotherapy. For this reason, parental cell line (231) and its derivatives resistant to doxorubicin (231**DXR**) and paclitaxel (231**PTR**) were used. The Mammosphere-Forming Assay and aldehyde dehydrogenase (ALDH) enzyme activity assay showed an increase in BCSCs in the doxorubicin-resistant model. Moreover, the expression of some transcription factors involved in epithelial-mesenchymal transition (EMT), a process that confers BCSC characteristics, was upregulated after chemotherapy treatment. FASN inhibitors C75, (−)-Epigallocatechin 3-gallate (EGCG), and its synthetic derivatives G28, G56 and G37 were used to evaluate the effect of FASN inhibition on the BCSC-enriched population in our cell lines. G28 showed a noticeable antiproliferative effect in adherent conditions and, interestingly, a high mammosphere-forming inhibition capacity in all cell models. Our preliminary results highlight the importance of studying FASN inhibitors for the treatment of TNBC patients, especially those who progress after chemotherapy.

## 1. Introduction

Breast cancer is the most common cancer among women worldwide with approximately 2.1 million diagnoses estimated in 2018 according to the International Agency for Research on Cancer [1]. Triple negative breast cancer (TNBC) represents approximately 15–20% of patients with breast carcinomas and is characterized by the lack of expression of estrogen and progesterone receptors (ER/PR) and neither expression nor amplification of the HER2 oncogene [2]. The absence of these three biomarkers prevents the use of currently available targeted therapies for breast cancer [3,4], and leaves systemic cytotoxic chemotherapy as the sole treatment option [5]. Despite a good initial response to chemotherapy in neoadjuvant settings, only 30% of patients will exhibit a survival of more than five years following diagnosis [6,7,8]. Compared to other BC subtypes, TNBC presents the highest recurrence rate along with the shortest time, which is associated with a significantly poorer overall survival [2,7]. Two major molecular subtypes are represented in TNBC, basal-like (80%) and claudin-low (CL) or mesenchymal-like (ML; 20%), this last one being enriched in cells with stem features [9]. Current therapies failure and resistance may be explained in part by the presence of a small population of cells that display stem properties within breast tumors known as breast cancer stem cells (BCSCs). In vitro, BCSC-enriched populations display specific features: cell-surface specific marker pattern (CD44^high^/CD24^low^), the ability to form mammospheres when growing in non-adherent conditions, aldehyde dehydrogenase 1 (ALDH1) enzyme activation, and enhanced resistance to chemotherapy [10]. Additionally, the induction of epithelial-mesenchymal transition (EMT) through *E-cadherin* expression decrease and upregulation of mesenchymal proteins, such as *vimentin* or *N-cadherin*, results in stem properties [11]. Some transcription factors involved in EMT regulation are *slug*, *zeb1*, *zeb2*, *twist* and *snail* [12,13].

Moreover, it has recently been demonstrated that the regulation of lipid metabolism promotes BCSCs and cancer chemoresistance [14]. Back in 1924, Warburg made evident metabolism deregulation in cancer cells [15,16], becoming many years later a hallmark of cancer [17]. Cell membranes are formed by long-chain fatty acids, being also important substrates for energy cell metabolism. The Fatty Acid Synthase (FASN) is the enzyme responsible for the de novo synthesis of palmitate, the most abundant fatty acid [18]. Several carcinomas such as breast, colon, lung, prostate, among others, overexpress FASN [19,20,21,22], suggesting it as a unique onco target. Blocking FASN activity causes in vitro and in vivo anticancer activity by inhibiting tumor progression [23,24,25,26,27,28], hindering angiogenesis [29,30], overcoming drug-resistance [31,32], and synergistically increasing the efficacy of chemotherapy [26,33,34]. A recent study showed that FASN was expressed in 92% of tumor tissue samples coming from a cohort of 100 TNBC patients and its association with positive node status made evident its role as a possible predictive biomarker in this aggressive BC subtype [35].

(−)-Epigallocatechin 3-gallate (EGCG) is a powerful antioxidant and the most abundant catechin in green tea. Its apoptotic effect leads to antiproliferative activity [36,37,38,39]. Although EGCG targets HER1-HER2, MAPK, and AKT signaling pathways among others, it has been described that its apoptosis-inducing effect occurs through FASN inhibition [28,40,41]. Several studies have demonstrated a weak effect of EGCG in 20 different human cancer stem cell populations when used alone but synergistically increased in combination with different anticancer drugs [42]. We have produced a battery of new polyphenolic derivatives structurally related to EGCG, from which G28, G56, and G37 proved to possess enhanced FASN inhibitory activity [43,44,45]. These compounds also showed cancer cell cytotoxicity in a set of human breast cancer cells. G28 displayed a potent tumor volume reduction in vivo with no weight loss or anorexia, the main side-effects of other FASN inhibitors like the cerulenin-derived compound C75 [28,41,43]. G28 also showed apoptosis induction in HER2+ resistant cell lines and tumor diminishment in HER2+ breast cancer xenografts [26,46].

In the present study, we evaluated FASN and BCSC characteristics, i.e., mammosphere-forming capacity and ALDH1 activity, in the acquisition of chemoresistance in the TNBC model MDA-MB-231 (231). Moreover, we used the natural FASN inhibitor EGCG and its synthetic derivatives G28, G56, and G37 in comparison to C75 (Figure 1) to target FASN through these BCSC features from these TNBC models resistant to doxorubicin (231**DXR**) and paclitaxel (231**PTR**), the most common drugs currently used in this BC subtype devoid of a validated targeted therapy.

## 2. Results

### 2.1. FASN Expression in MDA-MB-231 Derived Chemoresistant Cell Lines

FASN activity has demonstrated to play an important role in drug resistance through new phospholipid synthesis for membrane renovation and plasticity. It also decreases ceramide levels, inhibiting apoptosis via PARP activation [32,47,48,49,50].

To assess the role of FASN in chemoresistance acquisition in TNBC, we developed MDA-MB-231 (231) cells resistant to doxorubicin (231**DXR**) [34] and paclitaxel (231**PTR**) (Appendix A). It has been described that doxorubicin-resistant cell lines become sensitive through the inhibition of FASN [34,51]. Therefore, we studied how FASN protein levels were modified after drug treatment of sensitive and chemoresistant TNBC cells. Our results showed that 231**DXR** FASN levels experienced a 2-fold increase after 24 h of doxorubicin treatment (Figure 2A), while such effect was not observed in parental cells. On the other hand, paclitaxel did not show any effect on FASN protein levels neither in 231 nor in 231**PTR** (Figure 2B). PARP cleavage, a marker for apoptosis, is increased in parental cell lines compared to its resistant counterparts for both drugs, making evident their chemoresistance (Figure 2).

### 2.2. BCSC-Enriched Population in Sensitive and Resistant Cell Lines

Despite the fact that BCSCs are a very low-represented population within tumors, selection or enrichment of these cells after chemotherapy treatment can result in a tumor relapse due to their intrinsic chemoresistance and their tumorigenic ability [52,53,54]. Here, we aim at the evaluation of the BCSC population in our sensitive and resistant models by means of two of these techniques, i.e., mammospheres-forming capacity and ALDH1 activity. The influence of doxorubicin and paclitaxel treatment was also evaluated.

To determine whether there was an enrichment of this population in our resistant models, we used the Mammosphere-Forming Assay (MFA; Figure 3A) in 231 parental cell line and in the chemoresistant derivatives 231**DXR** and 231**PTR** (Figure 3B). 231 and 231**DXR** displayed similar Mammosphere-Forming Index (MFI), comparable to those of 231 and 231**PTR**. Likewise, no morphological differences were found among the three cell models (data not shown). Since one of the BCSC population characteristics is its inherent chemoresistance [55], we carried out doxorubicin and paclitaxel treatments in adherent and mammosphere cell culture, in both sensitive and resistant cell models. Cell density, chemotherapy drug dosage (concentration that inhibited 30% (IC_30_) of cell viability for doxorubicin (70 nM) or paclitaxel (5 nM) at 48 h), and treatment duration were performed equally in both approaches. The 3-(4,5-dimethylthiazol-2-yl)-2,5-diphenyltetrazolium bromide (MTT) assay was carried out in adherent conditions to determine cell viability, whereas, in non-adherent culture, the Mammosphere-Forming Inhibition (MFIn) was used. The BCSC-enriched population showed inherent doxorubicin resistance when compared to adherent conditions for both sensitive and resistant cell lines (Figure 3C). In adherent settings, the inhibition of viability ranged from 89.81% ± 1.65 (231) to 86.7% ± 1.85 (231**DXR**). Nevertheless, MFIn values exhibited lower inhibition in both cell lines treated with doxorubicin. Furthermore, in non-adherent culture, a doxorubicin cytotoxic effect was significantly higher in parental cells (60.66% ± 1.17) compared to the resistant model (51.29% ± 0.58, *p* < 0.01). Similar results were found with paclitaxel treatment (Figure 3D). The reduction in viability in adherent condition was significantly higher in sensitive cells (90.47% ± 1.40) compared with paclitaxel-resistant model (72.61% ± 0.94) with *p* < 0.001. Additionally, BCSC-enriched population in mammospheres assay showed paclitaxel intrinsic resistance, with MFIn values of 47.82% ± 3.17 (231) and 31.24% ± 5.59 (231**PTR**).

We next measured the ALDH1 activity using the ALDEFLUOR^TM^ assay. Obtained data revealed a significant increase of ALDH1 activity in the resistant counterpart 231**DXR** (26.52% ± 1.81) compared to parental (10.81% ± 5.74; *p* < 0.05) (Figure 3E). Interestingly, ALDH-bright (ALDH^br^) cells from both sensitive and resistant cell lines were also increased, with no significance, after 12 and 24 h of doxorubicin treatment, reaching values of 42.15% ± 10.53 and 38.10% ± 6.71, respectively. 231**DXR** cells exhibited a non-significant increase of ALDH1 activity in comparison with 231 cells after 12 h treatment (36.87% ± 3.56 vs. 17.28% ± 5.68), showing similar levels at 24 h after treatment.

Regarding an ALDH^br^ population in paclitaxel-resistant 231 cell line, no significant differences were found between sensitive and resistant cell models (Figure 3F). Paclitaxel treatment decreased ALDH^br^ population after 24 h of treatment in sensitive cells (*p* < 0.01), whereas ALDH^br^ population in 231**PTR** cells remained similar during treatment.

### 2.3. Evaluation of EMT in Drug-Resistant TNBC Models

Cell plasticity is a key process in the development of drug resistance [56]. It has been described that induction of EMT can lead to cell dedifferentiation, acquiring BCSC features such as chemoresistance [11,39]. We wondered if the development of this phenotype was due to EMT process triggered by chemotherapy treatment. To that purpose, RNA expression of the EMT-related transcription factors *snail*, *slug*, *twist*, *zeb1* and *zeb2* was evaluated after doxorubicin or paclitaxel treatment. The basal levels of *snail* were 2-fold higher in 231**DXR** compared to 231 (*p* < 0.05). When doxorubicin was added, interestingly, a significant increase of *snail* was observed in the 231 cell line, while such a strong effect was not observed in 231**DXR** (Figure 4A). *Vimentin* gene expression levels were analyzed since it is an intermediate filament protein related to the mesenchymal phenotype. After 12 h of doxorubicin treatment, *vimentin* expression was significantly increased in both cell lines with a higher growth in 231**DXR** (Figure 4A).

Concerning paclitaxel resistant cells, *slug* transcription factor was upregulated in 231**PTR** compared with sensitive 231 cells (*p* < 0.05) but opposite results were observed in *vimentin* levels at the same time points (Figure 4B).

### 2.4. C75, EGCG and Its Derivatives G28, G56, and G37 Effect in BCSC-Enriched Populations

MFIn was performed in 231, 231**DXR**, and 231**PTR** cell lines in order to study the impact of FASN inhibition in the BCSC-enriched population.

Five different FASN inhibitors i.e., C75, EGCG, and its synthetic derivatives G28, G56, and G37 [45], were used to evaluate their efficacy in reducing mammosphere-forming capability. The concentration that inhibited 30% (IC_30_) of cell viability at 48 h was used (Appendix A) [34]. Then, adherent and non-adherent cell cultures were treated with IC_30_ of each compound (Figure 5).

No differences were found with C75 treatment in adherent and non-adherent conditions in 231 or both resistant cell models (231**DXR** and 231**PTR**), with values ranging from 45.26% ± 2.94 to 21.24% ± 5.18. EGCG cytotoxic effect was significantly diminished in non-adherent settings in all cell lines. EGCG reduced viability in adherent cells with percentages ranging from 81.54% ± 3.38 (231) to 85.43% ± 0.99 in 231**PTR**. Instead, MFIn values laid between 25.71% ± 4.42 (231) and 36% ± 14.70 in 231**DXR**.

G28 displayed a noticeable cytotoxic effect in both adherent and suspension conditions. Although the reduction in viability ranged from 78.70% ± 2.50 (231) to 72.66% ± 6.76 (231**DXR**) and 83.50% ± 2.08 (231**PTR**) in adherent conditions, the same dose of G28 exhibited a significantly cytotoxic effect in suspension, with values of 98.10% ± 0.72 for 231, 94.70% ± 1.55 for 231**DXR** and 90.67% ± 1.60 for 231**PTR**.

Regarding G56 and G37 compounds, both exhibited a strong effect in adherent conditions in all cell models. For instance, cells treated with G56 showed a reduction in viability of 88.56% ± 0.57 (231), 79.31% ± 5.18 (231**DXR**) and 84.66% ± 1.63 (231**PTR**). As a counterpart, G37 treatment resulted in similar values, concretely 75.24% ± 5.32 for sensitive 231 cells, 81.33% ± 4.04 for doxorubicin-resistant cells and 83.07% ± 1.93 for 231**PTR** cells. On the other hand, both compounds showed lower MFIn effect, especially G56, which exhibits decreased values of 27.56% ± 7.20 (231), 28.01% ± 3.42 (231**DXR**), and 28.47% ± 6.86 (231**PTR**), similar to those found in EGCG treatment.

## 3. Discussion

The two major molecular subtypes within TNBCs are the CL and ML. The last one accounts for 20% of TNBCs (the second in incidence after basal-like in TNBC [9]), and exhibits an enrichment of the EMT features, showing BCSC properties, and having a poor prognosis [9,57]. Furthermore, the enrichment of CSC features has been found in tumors that progressed after therapy [52,53]. In the present study, an ML subtype cell line MDA-MB-231 (231) and its resistant derivatives 231**DXR** and 231**PTR** were used to evaluate the BCSC population in chemotherapy resistance acquirement.

BCSCs are a low-represented population within tumors, primary and established cell lines; therefore, different methodologies have been settled to study the BCSC-enriched population with these cells in vitro. The 231 cell line is known to form vague mammospheres that can be passaged for several generations that can be passed on for several generations [58,59,60,61]. While no differences were found regarding the index formation between the three cell lines assayed, significant differences were observed using this methodology under drug treatment. Both 231**DXR** and 231**PTR** showed increased ability to form mammospheres under drug treatment compared to 231 showing both, drug resistance and anchorage-independent growth. The ALDEFLUOR^TM^ assay indicated that ALDH^br^ cell population was significantly increased in the 231**DXR** cell line, whereas ALDH^br^ 231**PTR** cell percentage remained similar to that in the parental. Interestingly, the number of ALDH^br^ cells significantly increased in the parental cell line after doxorubicin treatment. Taken together, these results suggest that 231**DXR** cell line shows an increase in BCSC features, and that these features can be induced after doxorubicin treatment. On the other hand, despite the fact that we did not observe an obvious enrichment in BCSC characteristics in 231**PTR** compared to parental cells, the intrinsic BCSC features of this ML cell line were maintained.

Cell plasticity plays a key role in tumor progression, and the plastic CSC concept defines that bidirectional conversions between non-CSCs and CSCs may exist. Thus, the activation of the EMT process may lead to the CSC phenotype [11,38,39]. 231**DXR** and 231**PTR** cell lines showed higher resistance to their respective chemotherapeutic agent in comparison with 231 in adherent or suspension condition. Additionally, and as expected, doxorubicin and paclitaxel showed more cytotoxic effects in adherent conditions compared to non-adherent ones. Interestingly, the ALDH1 activity was enhanced after doxorubicin treatment in both sensitive and resistant 231**DXR** cell models. This phenotype is associated with chemoresistance in patients [62]. Remarkably, basal expression of *snail*, an EMT inducer directly associated with tumor development and relapse [63,64,65], significantly increased in 231**DXR** compared to sensitive cells. Regarding paclitaxel resistance, no differences were observed in the ALDH^br^ population. The EMT-related transcription factor *slug*, linked to cancer progression and BCSC activity [66], was upregulated in 231**PTR** cells compared to 231 cell model with no changes in *vimentin* levels.

Although doxorubicin counts with undeniable advantages as an antitumor drug, it has been shown to promote stemness in murine cell lines [67,68,69]. Recent evidence showed that doxorubicin treatment increased stem cell-related signaling pathways, explaining the inherent chemoresistance of MDA-MB-231 (compared to another ML cell line, Hs578T) [70]. Previous studies showed the ability of paclitaxel to increase the BCSC subpopulation in 231 TNBC cell lines [71]; however, non-obvious enrichment was observed in our hands, but a maintenance of the intrinsic stem features of this ML subtype.

FASN overexpression confers many advantages to tumor cells such as the ability to preserve a high proliferation rate, and it also plays a key role in drug resistance acquisition [32,47,49]. We have proven that FASN inhibition overcame doxorubicin resistance in chemoresistant cell lines [34], and FASN protein levels significantly increased in the 231**DXR** cell model when treated with doxorubicin. It has been previously described in breast cancer that FASN protein levels increase as a mechanism to become resistant [72]. Gonzalez-Guerrico and coworkers showed that the inhibition of FASN led to the conversion of the ML phenotype to a non-malignant one by downregulating EMT markers in a breast cancer model [73]. Additionally, the FASN inhibitor metformin blocked lipogenesis, leading to a switch from basal- to luminal-like sphere morphologies [73]. Likewise, resveratrol, another FASN inhibitor, precluded the growth of the CSC population both in vitro and in vivo using a breast cancer xenograft model [74]. FASN expression has also been shown to be important in stemness preservation in glioma [75]. In this work, we evaluated the effect of the inhibition of FASN in an CSC population through the ability to hinder mammosphere-formation capacity compared to the cytotoxic effect in adherent culture. No differences were found using C75 in adherent or suspension conditions. G28 displayed a noticeably better inhibitory effect in non-adherent conditions in both sensitive and resistant cell lines. Interestingly, this outcome was not found when using any of the chemotherapeutical agents (doxorubicin or paclitaxel). These results set up the basis for further investigation using FASN inhibitors in combination with chemotherapy to target BCSCs as well as the bulk of the tumor to improve the outcome of patients with TNBC.

In conclusion, the results presented in this work suggest that chemoresistance of 231**DXR** and 231**PTR** (ML cell models) is not only a static drug-selected state, but also an acquired reversible phenotype that can be triggered during treatment. 231**DXR** presented a larger BCSC population compared to the doxorubicin sensitive 231 cell line, which was also enhanced with the presence of the chemotherapeutical agent. This might be explained by the activation of EMT induced through *snail*. Despite the preliminary nature of the obtained results, they provide a rationale to suppress FASN as a potential strategy for a highly proliferative neoplasia such as TNBC.

## 4. Materials and Methods

### 4.1. Cell Culture and Development of Doxorubicin- and Paclitaxel-Resistant TNBC Cells

MDA-MB-231 cell line was obtained from the American Type Culture Collection (ATCC) and was routinely grown in DMEM (Gibco, Thermo Fisher Scientific, Waltham, MA, USA) supplemented with 10% FBS (HyClone Laboratories, GE Healthcare, Chicago, IL, USA), 1% l-glutamine (Gibco), and 1% sodium pyruvate (Gibco), 50 U/mL Pen/Strep (Linus). Cells were kept at 37 °C and 5% CO_2_ atmosphere. Doxorubicin-resistant cells MDA-MB-231 (231**DXR**) and paclitaxel-resistant cells (231**PTR**) were developed using a stepwise selection method. Increasing doses of doxorubicin (TEDEC-Meiji Farma, Alcalá de Henares, Spain) and paclitaxel (Accord Healthcare Ltd., Thaltej, India) were added until their corresponding IC_50_ was reached. Briefly, cells were initially treated with a concentration of 0.1xIC_50_ of each drug. After 48 h, treatment medium was replaced for fresh medium. When the cells were capable of growing and reached appropriate confluence, they were treated with double the previous concentration for 48 h. The stepwise selection method was subsequently performed until the final concentration of drug was that of the parental IC_50_. It took around six months for each cell line. Resistance was confirmed by cell viability assay.

### 4.2. Western Blot Analysis of Cell Lysates

Cell lines were treated during 12 or 24 h with half the concentration of doxorubicin that inhibited 50% (0.5xIC_50_) of the MDA-MB-231 parental cells’ viability previously determined with MTT assay after 48 h [34]. A final concentration of 3xIC_50_ of paclitaxel was used (Appendix A). After treatment, parental and resistant TNBC cells were lysed in ice-cold lysis buffer (Cell Signaling Technology Inc., Danvers, MA, USA) with 100 μg/mL phenylmethylsulfonyl fluoride (PMSF) by vortexing every 5 min for 30 min. Equal amounts of protein were heated in an lithium dodecyl sulfate (LDS) Sample Buffer with Sample Reducing Agent (Invitrogen, Carlsbad, CA, USA) for 10 min at 70 °C, electrophoresed on SDS-polyacrylamide gel (SDS-PAGE), and transferred onto nitrocellulose membranes. Blots were incubated for 1 h in blocking buffer (5% powdered-skim milk in Phosphate-buffered saline 0.05% Tween (PBS-T)) and incubated overnight at 4 °C with the appropriate antibodies diluted in blocking buffer (Table 1). Specific horseradish peroxidase (HRP)-conjugated secondary antibodies were incubated for 1 h at room temperature. The immune complexes were detected using a chemiluminescent HPR substrate (Super Signal West Femto (Thermo Scientific Inc.) or Immobilon Western (Merck Millipore, Burlington, MA, USA)). β-actin (Santa Cruz Biotechnology Inc., Dallas, TX, USA) was used as a control of protein loading. Western blot analyses were repeated at least four times and representative results are shown. Quantification analyses of Western blot data were performed using a computer-assisted densitometer. The results are expressed as ratio of protein levels vs. β-actin levels.

### 4.3. Cell Viability Assays

To elucidate inhibitory effects of chemotherapy drugs and FASN inhibitors, 231 and resistant counterparts were seeded in 96-well microplates at a cell density of 4 × 10^3^ cells/well in their corresponding growth medium. After 24 h, culture medium was removed and 100 µL of fresh medium with increasing doses of paclitaxel, or FASN inhibitors (EGCG (Sigma-Aldrich, St. Louis, MO, USA; dissolved in PBS 5% DMSO), C75 (Sigma), EGCG-derivatives G28, G56 or G37 (dissolved in DMSO)) were added to each well. The synthesis of FASN inhibitors G28, G56, and G37 was performed as previously described [45]. Following 48 h treatment, a colorimetric MTT assay was used to measure cell viability as previously described [27]. Graph plots were performed in order to elucidate paclitaxel resistance and inhibitory concentrations.

On the other hand, cell viability inhibition experiments were performed seeding 5000 cells in adherent 6-well cell culture microplates with DMEM. Cells were incubated for 24 h to allow cell attachment and then chemotherapy drugs (doxorubicin or paclitaxel) or FASN inhibitors were added at a unique specific concentration (IC_30_ of parental cells calculated at 48 h) for five days. Finally, cell viability was also measured using a colorimetric MTT assay.

### 4.4. Mammosphere-Forming Assay

The mammosphere-forming procedure was used to evaluate an CSC population, following previously described protocols [55,56]. After cell detachment, 2000 cells were seeded into a non-adherent 6-well cell culture microplate. Cells were cultured with DMEM/F12 medium supplemented with B27, hEGF and hFGF (20 ng/mL), 1% l-glutamine, and 1% sodium pyruvate. Microplates were incubated for 5 days and mammospheres bigger than 50 µm were counted using an inverted optical microscope. Different parameters were calculated using the formulas described above: (**A**) the Mammosphere-Forming Index (MFI) and (**B**) Mammosphere-Forming Inhibition (MFIn). For MFIn analysis, compounds were added at the moment of seeding. An IC_30_ of chemotherapy drugs and FASN inhibitors calculated at 48 h on MDA-MB-231 parental cells was used.

Formula: 
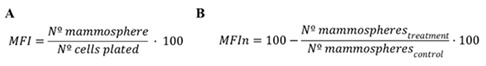


### 4.5. Quantitative Real-Time PCR Analysis

Cells were treated with a concentration of doxorubicin or paclitaxel equal to the 0.5xIC_50_ and 3xIC_50_, respectively, obtained after 48 h of treatment. After 12 or 24 h treatment, cells were washed with PBS, and then Qiazol (Qiagen, Hilden, Germany) was added. RNeasy mini kit (Qiagen) was used to isolate total RNA according to manufacturer instructions. RNA was reverse-transcribed into complementary DNA (cDNA) using High Capacity cDNA Archive Kit (Applied Biosystems, Foster City, CA, USA). Different gene expression levels were determined using LightCycler^®^ 480 Real-time PCR System (Roche, Basel, Switzerland) with LightCycler^®^ 480 SYBR Green I Master (Roche), following manufacturer instructions. Primers used are shown in Table 2. RT-PCR analyses were performed at least three times and each gene was run in triplicate. Gene expression levels were quantified using the standard formula 2^ΔCT and normalized to the housekeeping gene β-actin (2^ΔΔCT).

### 4.6. Aldefluor Assay

Aldefluor^TM^ kit (STEMCELL Technologies) was used to determine ALDH enzyme activity, following company guidelines. Briefly, ALDH reagent (BAAA) freely diffuses into viable cells and ALDH enzyme catalyzes its conversion to BAA, which is negatively charged and retained into cells. Intracellular BAA increases fluorescence which can be analyzed through flow cytometry. As seen before for 12 and 24 h treatments, 0.5xIC_50_ and 3xIC_50_ (calculate at 48 h on parental cells) for doxorubicin and paclitaxel, respectively, were used. After treatment, cells were detached and 2 × 10^5^ cells per sample were collected. After washing with PBS, 500 µL of the kit buffer was added to each condition. Afterwards, 2.5 µL of the ALDH reagent was added and 250 µL of the suspension was immediately transferred to a new Eppendorf with 2.5 µL of the ALDH inhibitor *N*,*N*-diethylaminobenzaldehyde (DEAB), in order to consider background fluorescence (Appendix A). All samples were incubated at 37 °C for 40 min. Cells were then centrifuged and washed and samples were prepared to be analyzed through cytometry (FACSCalibur II, BD Bioscience). Images were obtained with FlowJo software (Flow Jo LLC, Ashland, OR, USA.

## 5. Conclusions

The study of the BCSC-enriched population in an ML TNBC cell line (MDA-MB-231) and its doxorubicin-resistant derivative (231**DXR**) showed increased BCSC features in both sensitive and resistant cell models after chemotherapy treatment. We have previously recognized FASN as a possible co-target to resensitize chemoresistant cells to doxorubicin [34]. Moreover, its implication in drug resistance acquisition in breast cancer has already been identified [72]. Results derived from the present paper showed that the newly developed FASN inhibitor EGCG-derived G28 had a strong inhibitory effect on the mammosphere-formation capacity (a stem feature) in sensitive and both drug-resistant models, 231**DXR** and 231**PTR**. However, these outcomes are still in initial states and more approaches should be performed. According to recent published research, FASN would not only play an important role in a highly proliferative neoplasia such as TNBC, but also in drug resistance acquisition and in sustaining malignancy in cancer [73,74,75], being a promising (co)target for patients with TNBC that progress to current treatments.

## Figures and Tables

**Figure 1 molecules-24-01027-f001:**
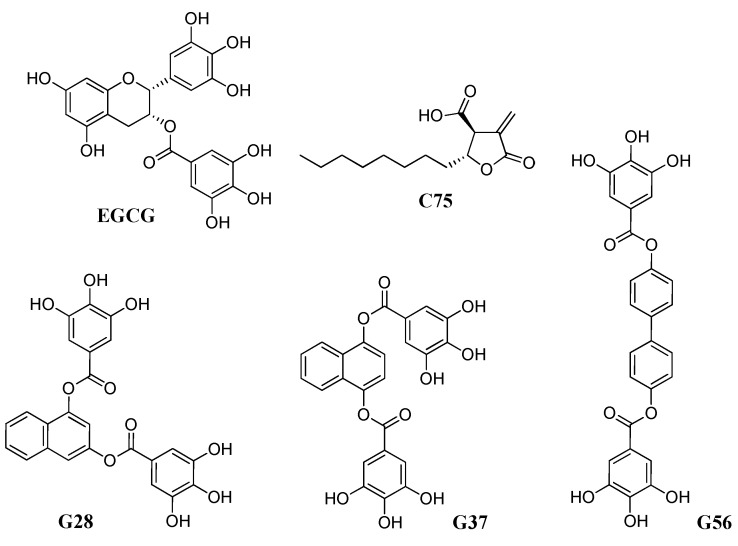
Structure of compounds EGCG, C75, G28, G37, and G56.

**Figure 2 molecules-24-01027-f002:**
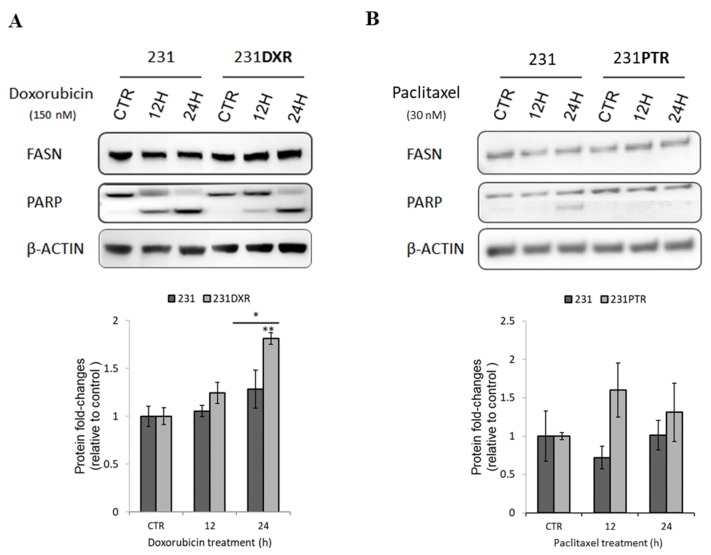
FASN protein expression after doxorubicin or paclitaxel treatment in 231, 231**DXR** or 231**PTR** cell lines. Western blot analysis of FASN, and PARP expression after 12 h or 24 h of (**A**) doxorubicin (150 nM) or (**B**) paclitaxel (30 nM) treatment. FASN levels are normalized by β-actin and expressed as fold-changes relative to control at each time point. Experiments were performed four times. *(*p* < 0.05) and **(*p* < 0.01) indicate levels of statistically significance.

**Figure 3 molecules-24-01027-f003:**
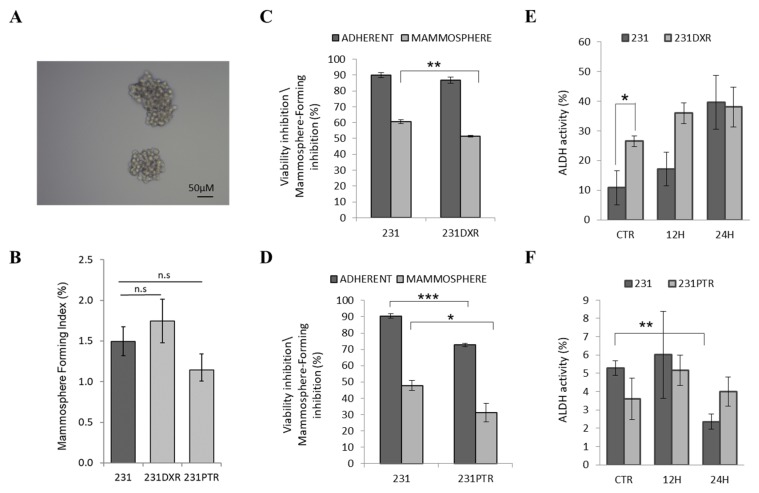
Mammosphere-Forming Assay and ALDH activity assay in resistant cells. (**A**) Representative image of mammospheres from 231 (parental) cell line at day 5; (**B**) Mammosphere-Forming Index (MFI) for 231, 231**DXR** and 231**PTR**. Cell viability inhibition and Mammosphere-Forming Inhibition (MFIn) for (**C**) doxorubicin (70 nM) in 231 and 231**DXR** or (**D**) paclitaxel treatment (5 nM) in 231 and 231**PTR** for five days. In the same conditions, ALDH activity quantification (%) for (**E**) 231**DXR** and (**F**) 231**PTR** compared to 231. Results are expressed as mean ± SEM. * (*p* < 0.05), ** (*p* < 0.01) and *** (*p* < 0.001) indicate levels of statistically significance or n.s. when no significance was found.

**Figure 4 molecules-24-01027-f004:**
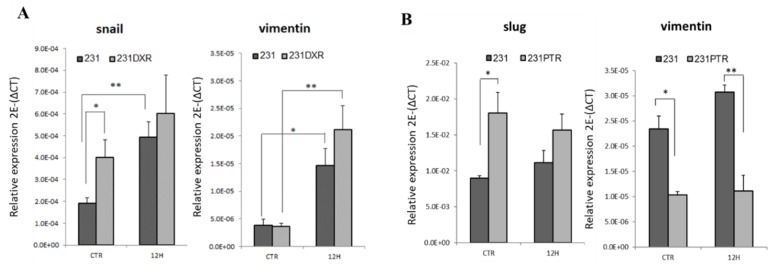
EMT-related genes expression after chemotherapy agents’ treatment. (**A**) *snail* and *vimentin* gene expression in 231 and 231**DXR** cells after 12 h of doxorubicin treatment (150 nM); (**B**) *slug* and *vimentin* gene expression in 231 and 231**PTR** after 12 h of paclitaxel treatment (30 nM). Experiments were performed at least three times. * (*p* < 0.05), ** (*p* < 0.01) and *** (*p* < 0.001) indicate levels of statistically significance.

**Figure 5 molecules-24-01027-f005:**
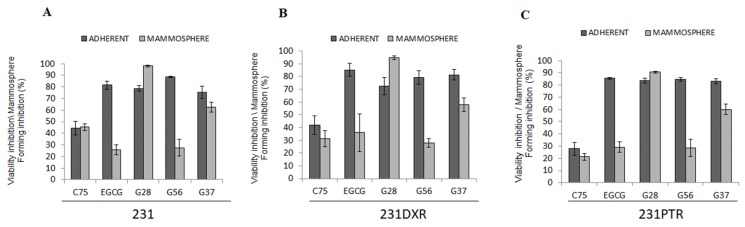
Cell viability inhibition vs. Mammposphere-Forming Inhibition (MFIn) for FASN inhibitors EGCG, C75, G28, G56 and G37. (**A**) 231, (**B**) 231**DXR**, and (**C**) 231**PTR** were treated for five days with C75 (30 µM), EGCG (120 µM), G28 (50 µM), G56 (50 µM), and G37 (50 µM). Experiments were performed at least three times in duplicate.

**Table 1 molecules-24-01027-t001:** Antibody description.

Antibody	Reference	Supplier	Dilution	Source
FASN	ADI-905-069-100	EnzoLife Sciences	1:1500	rabbit
PARP	9542	Cell Signaling Technology	1:1000	rabbit
β-actin	Sc-47778	Santa Cruz Inc.	1:1000	mouse

**Table 2 molecules-24-01027-t002:** Primer design.

FASN	Forward	CAGGCACACACGATGGAC
	Reverse	CGGAGTGAATCTGGGTTGAT
Snail	Forward	GCTGCAGGACTCTAATCCAGA
	Reverse	ATCTCCGGAGGTGGGATG
Vimentin	Forward	TGGTCTAACGGTTTCCCCTA
	Reverse	GACCTCGGAGCGAGAGTG
β-actin	Forward	ATTGGCAATGAGCGGTTC
	Reverse	CGTGGATGCCACAGGACT

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
