# Peer review of "EGCG-Derivative G28 Shows High Efficacy Inhibiting the Mammosphere-Forming Capacity of Sensitive and Resistant TNBC Models"

_molecules, 2019, doi:10.3390/molecules24061027_

Round 1

Reviewer 1 Report

The manuscript molecules-437119-peer-review-v1 by Giró-Perafita A et al., studied the effect of Epigallocatechin 3-gallate (EGCG), and its synthetic derivatives on the

The expression of fatty Acid Synthase (FASN) and cancer stem cell (CSC) phenotype in MDA-MB-231 cell line and  its chemo resistant derivatives as a model for triple negative breast cancer (TNBC). The authors used MDA-MB-231 cells generate chemoresistant cells after treatment with doxorubicin (231DXR) and paclitaxel (231PTR).  They found EGCG- derivative G28 inhibits proliferation and  mammosphere formation in all cell models. However, I have some concerns need to be clarified to make the manuscript possibly accepted.

Major Concerns:

1.     On which basis the authors used a concentration of 0.1xIC50  for each drug. Please clarify?

2.     On which basis the authors used concentration of 150 nM of doxorubicin and 30 nM of paclitaxel in their experiments. Please clarify?

3.     In Figure 2, if the authors set control to 1 value, how there is SD?

4.     It would be great if the authors provide a panel for mammoshere formation image.

5.     Figure 3F is of bad resolution. Please replace.

6.     It is not clear how the authors calculate gene expression level in Figure 4. Please clarify in detail in method section.

7.     It would be helpful to verify one or 2 of EMT markers on protin levels by Western blot.

8.     Why the effect of EGCG and C75 on viability of 231DXR and 231PTR cells are not shown in Supp. Figure 2.

9.     Why IC3 used not IC50 in FASN inhibition experiments.

10.  Where are results that confirm the inhibitory effect of EGCG- derivatives on FASN expression?

11.  Effect of G28 on ALDH- activity can be tested.

12.  Molecular mechanism should be addressed to identify how G28 affect FASN expression.

Minor Concerns

1. Title

BCSC should be fully named

2. Abstract

 FASN inhibitors C75, (-)-Epigallocatechin 3-gallate (EGCG), and its synthetic derivatives G28, G56 and G37 were used to evaluate the effect of FASN inhibition in the BCSC-enriched population in our cell lines. In should be changed into on.

3. In result section

What is ALDHbr?

4. In method section

·      500mL of the kit buffer was added? How 500mL?

·      cycle program in qPCR should be written?

Overall. I recommend major revision.

Author Response

The reply to the reviewer is attached in a pdf file. 

Reviewer 2 Report

The issue is interesting but the paper suffers from some  flaws:

Introduction

This section is quite confused and contains some inaccuracies; 

-page 2, lines 47-48: "good initial reponse to chemotherapy in adjuvant setting", maybe authors intended "neoadjuvant"?

-page 2, lines 75-80: tis sentence is very confused

-page 2, lines 92-94 and page 3, lines 95-96: this last part of the Introduction section anticipates results and must be deleted

Results

This section should only describe the most important data without introduction/comments.

Instead, most paragraph contain sentences that must be placed in Introduction or Discussion sections (for example: page 3, lines 101-108; page 4, lines 120-128; page 6, lines 171-175)

Discussion

This section contains some confused sentences/paragraphs, for example:

Page 9, lines 273-283

Conclusions

This paragraph must be shortened;  it should only  summarize  the principal findings of the study and future implications/directions

I suggest authors to appropriately  re-organize their paper 

Some English mistakes must be corrected

Author Response

(The authors gave the same response as above.)

Reviewer 3 Report

Comments to the Authors:

The abbreviation ALDH has not been explained in the whole manuscript.

In the Introduction part: I have no objections in the field of Introduction

Methods:

1.       I would like to know if doxorubicin- and paclitaxel-resistant cells following selection have been cultured in “pure” DMEM or in the presence of a small, maintaining dose of these drugs? If the drugs were not present in the culture medium, the cells could again become more sensitive to these compounds. I have got the impression that the tested cells are not so much resistant to paclitaxel, since about 40% of the cells described as paclitaxel-resistant die following the cytostatic drug treatment at the concentration equal the IC50 for the control cells (Supplementary Figure 1). Could you explain that?

2.       What was the IC50 for doxorubicin treatment of parental MDA-MB-231 cells and how IC50 values were calculated in the case of both drugs?

3.       Please explain what is the difference between the MTT-based viability test and the proliferation test? The only differences between them described in the method section are that in the viability test, the cells were grown in a 96-well plate for 48h, while in the proliferation test the cells were grown in a 6-well plate for 5 days. I do not think that on this basis it is possible to distinguish the viability of cells under the influence of the drug from their proliferation.  Moreover, the data obtained by exactly the same method (MTT) are once described as inhibition of proliferation (Figure 3) and elsewhere as inhibition of viability (Figure 5).

4.       Please cite the source of the protocol used for execution of mammosphere-forming assay in method section.

5.       Please rewrite description of Aldefluor assay because the current one is incomprehensible – it should contain at least a brief description of the principle of the method. Moreover, the volumes given in the description are as I suspect incorrect. I believe that you have added 500µl instead of 500ml of buffer to cell suspension. Am I right?

Results:

My main objection to the presented results is connected to the fact that the Authors show statistical differences between the data obtained using two completely different methods (MTT assay and mammosphere -forming assay) - Figure 3 and 5. I think that we cannot analyze this type of data together. Authors can compare observed trends, however they shouldn’t assign them the statistical significance. In my opinion, the charts presenting the above-mentioned data should be separated and analyzed individually.

Other comments:

1.       Results presented in Figure 2:

·       On what basis did you choose used concentration of doxorubicin (150nM) and paclitaxel (30nM)? And why are they different from these used in experiments presented in Figure 3 (200nM for doxorubicin)?

·       Why did you put a picture of PARP protein bands in the Figure if you do not comment on it in the text?

·       Line 111 of the text- you shouldn’t write about upregulation if observed differences are not statistically significant.

2.       Results presented in Figure 3

·       Line 153 and line 158 of the text- you shouldn’t write about increased or decreased ALDH activity if results do not differ in a statistically significant way.

·       The part of the figure 3F is just the presentation of raw data included in part 3D and due to that should be transferred to supplementary materials.

3.       Results presented in Figure 4

·       Could you tell me why expression of EMT-related transcription factors was analyzed based on the RNA and not on the protein level?

·       Line 177 – increase of Snail expression in 231DXR cells occurring after incubation for 12 h with doxorubicin was not statistically significant.

4.       Results presented in Supplementary Figure 2

·       Why did you decide to use the concentration of tested compounds  that inhibited 30% of cell viability? How did you calculate the IC30 values and what are these values? Why in part A and B of the Figure the only cell line tested was 231PTR line?

5.       Results presented in Figure 5

·       In what kind of solvent the FASN inhibitors synthesized by the Authors are dissolved? In my opinion the chart should also include results for the control - cells treated only with the solvent which was used.

·       I would also like to know what volumes (in µl) of tested compounds were added to the cells grown in 96-well plate to achieve the IC30 concentration?

·       From the graphs presented in Figure 5 it can be deduced that the G28 inhibitor results in a similar reduction of proliferation in both - parental cell line and lines resistant to doxorubicin and paclitaxel, therefore it is not more effective against resistant cells. Could the Authors comment on this observation?

Discussion

1.       In the Lines 232-235 Authors wrote:No differences were found regarding the index formation through the MFA between none of the three cell lines assayed, neither sensitive nor resistant. The ALDEFLUORTM assay  however, indicated that ALDHbr cell population was significantly increased in the 231DXR cell line whereas 231PTR ALDHbr cell percentage remained similar to that in the parental. “ So generally line 231DXR presents only two of three hallmarks described as features of cancer stem cells (resistance to chemotherapy, increased ability to from mammospheres and enhanced ALDH activity), while line 231PTR demonstrates only one of these features. Moreover activation of the EMT process in 231PTR cells is not evident. The title of the manuscript reads as follows “EGCG-derivative G28 shows high efficacy in reducing BCSC-enriched population in TNBC models of acquired drug resistance” - so maybe it will be reasonable to exclude 231PTR line from this research according to the fact that it cannot be treated as a model of cells resistant to chemotherapy and enriched in BSBC population.   

2.       Lines 278-279: The Authors state:231PTR showed milder phenotype, but also an increase of the BCSC-enriched population after  paclitaxel treatment was observed. “ On what basis Authors draw this conclusion?

3.       Lines 281-283: The Authors state: “It also seems to revert the BCSC-acquired phenotype, overcoming resistance to chemotherapy, thus being a good candidate to be explored for patients that progress to first-line chemotherapy.” – taking into account preliminary character of the research I think that this statement is too strong.

4.       I am also wondering what effect these FASN inhibitors exert on normal cells that express this enzyme (e.g. adipocytes or liver cells) and what would be the side effects associated with the use of these drugs in patients. Could you comment on this?

Author Response

(The authors gave the same response as above.)

Round 2

Reviewer 1 Report

The manuscript has been improved. accepted in its present form

Author Response

We thank the reviewer observation. We have revised the manuscript correcting few mistakes in it. The revised version of the manuscript is attached with track of changes option. 

Reviewer 3 Report

Dear Authors,

You can find my remarks below (marked in red):

1. I would like to know if doxorubicin- and paclitaxel-resistant cells following selection have been cultured in “pure” DMEM or in the presence of a small, maintaining dose of these drugs? If the drugs were not present in the culture medium, the cells could again become more sensitive to these compounds. I have got the impression that the tested cells are not so much resistant to paclitaxel, since about 40% of the cells described as paclitaxel-resistant die following the cytostatic drug treatment at the concentration equal the IC50 for the control cells (Supplementary Figure 1). Could you explain that?

We developed our in vitro models of drug resistance with increasing doses of each drug (from low to high) until the IC50 of the parental cell line at 48h was reached. We ensure that cells were resistant at several passages after treatment by checking viability at increase doses of doxorubicin/paclitaxel. In addition to keeping cells in low passage, they have been cultured with low concentrations of the drug (10 nM for doxorubicin and 0.5 nM for paclitaxel).

How many times and after how many passages have you verified viability of cells in the presence of doxorubicin/paclitaxel? Were the results similar to these obtained just after resistant cell models generation? Could you present results of these tests?

Please clarify what does it mean „to keeping cells in low passage, they have been cultured with low concentrations of the drug (10 nM for doxorubicin and 0.5 nM for paclitaxel)”. Resistant cells were cultured in the presence of these very low doses of used drugs?

On what basis did you choose used concentration of doxorubicin (150nM) and paclitaxel (30nM)? And why are they different from these used in experiments presented in Figure 3 (200nM for doxorubicin)?

150 nM of doxoribin represents 0.5xIC50 and 30 nM of paclitaxel to 3xIC50. All the IC values were calculated at 48h using the MDA-MB-231 parental cell line through the MTT assay. For long-term treatment experiments i.e. mammosphere formation inhibition (section 2.2 and 2.5) an IC30 was used. On the other hand, for up to 24h treatment experiments 0.5xIC50 was used (section 2.1, 2.2 and 2.3). Nevertheless, with paclitaxel treatment a higher dose than 0.5xIC50 was required in short-term experiments, thus the real dose used was 3xIC50. The two drugs used have a different mechanism of action. Doxorubicin is an anthracycline that works as a DNA intercalator that impairs RNA synthesis, showing a long-term effect in cells even several days after treatment. Instead, paclitaxel, a taxane involved in microtubule destabilization during cell division, has a shorter effect on cell viability after is removed from cell culture.

I understand that you used doses representing 0.5xIC50 for doxoribin and 30 nM of paclitaxel what is equal 3xIC50. The question is – why did you choose exactly these concentrations? Did you earlier test many different concentrations and choose these? On what basis?

1. In the Lines 232-235 Authors wrote: “No differences were found regarding the index formation through the MFA between none of the three cell lines assayed, neither sensitive nor resistant. The ALDEFLUORTM assay however, indicated that ALDHbr cell population was significantly increased in the 231DXR cell line whereas 231PTR ALDHbr cell percentage remained similar to that in the parental. “ So generally line 231DXR presents only two of three hallmarks described as features of cancer stem cells (resistance to chemotherapy, increased ability to from mammospheres and enhanced ALDH activity), while line 231PTR demonstrates only one of these features. Moreover activation of the EMT process in 231PTR cells is not evident. The title of the manuscript reads as follows “EGCG-derivative G28 shows high efficacy in reducing BCSC-enriched population in TNBC models of acquired drug resistance” - so maybe it will be reasonable to exclude 231PTR line from this research according to the fact that it cannot be treated as a model of cells resistant to chemotherapy and enriched in BSBC population.

We believe the reviewer is partially right in this statement.

We think is interesting to keep the paclitaxel resistant model for different reasons. First, we were interested in testing FASN inhibitors in models of acquired resistance to the major chemoagents used in the clinic for the treatment of TNBC, doxorubicin and paclitaxel. We differ from the reviewer in considering paclitaxel cells not resistant. Viability curves and IC50 values are significantly higher.

We agree instead that this model, 231PTR, cannot be considered a proper model of resistance due to an enrichment of stem-cell features. We have improved the article to this direction, to give a clear idea that the cancer stem cell features are not enriched in the 231PTR model compared to the parental. But, we chose MDA-MB-231 cell line because has already been described to be enriched in cancer stem cell features, and this features, are present (or in some of the approaches tested increased) in the 231PTR cell line. Therefore, it makes this model valuable and interesting for the article.

In line with this, we changed the discussion as it does not reflect the general idea of the paper. It has been shown that the stemness acquired by the cancer cells can be a transient phenotype, triggered by the drug pressure that can activate EMT. That is why we kept our cells in treatment to test several stem cell features, and the reason we test EMT transcription factors activation. Therefore, we think is more valuable the results obtained in mammosphere forming assays under drug treatment because: 1. Show the intrinsic resistance of this CSC-enriched population to the drug, 2. This population is increased in all the models of drug resistance.

The changes might be found in the manuscript as we have used track of changes to this purpose.

Due to the fact that the Authors really want to present results for the 231PTR model, I would suggest to change the manuscript title - so that it does not suggest that both models described in the manuscript are characterized by increased number of BCSCs, e.g. “EGCG-derivative G28 shows high efficacy in reducing viability of TNBC models of acquired drug resistance”.

I have no objection to the rest of the corrections made by Authors.

Best regards

Author Response

We thank the reviewer's constructive criticism. Attached you will find the replies to the reviewer's concerns. 
